# A platform for blue-luminescent carbon-centered radicals

Xin Li[1], Yi-Lin Wang[1], Chan Chen[1], Yan-Yan Ren[1] & Ying-Feng Han [1] ✉

Organic radicals, which have unique doublet spin-configuration, provide an alternative method to overcome the efficiency limitation of organic light-emitting diodes (OLEDs) based on conventional fluorescent organic molecules. Further, they have made great breakthroughs in deep-red and near-infrared OLEDs. However, it is difficult to extend their fluorescence into a short-wavelength region because of the natural narrow bandgap of the organic radicals. Herein, we significantly expand the scope of luminescent radicals by showing a new platform of carbon-centered radicals derived from N-heterocyclic carbenes that produce blue to green emissions (444–529 nm). Time-dependent density functional theory calculations and experimental investigations disclose that the fluorescence originates from the high-energy excited states to the ground state, demonstrating an anti-Kasha behavior. The present work provides an efficient and modular approach toward a library of carbon-centered radicals that feature anti-Kasha's rule emission, rendering them as potential new emitters in the short-wavelength region.

Organic radicals are compounds that feature an unpaired electron with high reactivity. They have been recognized as reactive intermediates and play important roles in many chemical and biological systems[1]. This field has undergone a profound revolution and has been extensively investigated as building blocks for magnetic, conductive, and optoelectronic functional materials[2–5]. Unfortunately, most organic radicals are weakly luminescent or non-luminescent because of their strong non-radiative energy relaxation pathways[6]. Since the first doublet-emission organic light-emitting diodes (OLEDs) was developed[7], the field has attracted considerable attention in recent years. Luminescent organic radicals as emitters for OLEDs can theoretically achieve 100% internal quantum efficiency because of their characteristic doublet emission; thus, the external quantum efficiency of photoelectric devices can be improved[8–10]. Thus far, the reported organic radicals with inherent fluorescent are mainly derived from triaryl methyl fragments (Fig. 1a)[11–18]. In these radicals, fluorescence produces radiative transition from the lowest excited state ($D_1$) to the ground state ($D_0$). According to Kasha's rule, such radicals have advantages in red and near-infrared (NIR) emissions because of their natural narrow bandgap (Fig. 1b)[8]. New categories of organic radicals with short-wavelength emission (particularly blue emission) should be

researched, they can be applied to further develop luminescent radicals and radical-based OLEDs.

In search of a new platform for isolating functionalized carbon-centered radicals, we report here the synthesis and structural study of a series of luminescent carbon-centered radicals by incorporating diphenylaminophenyl (TPA) and carbazolylphenyl (CBP) to the core of N-heterocyclic carbenes (NHCs) (Fig. 1c). The targeted NHC-based radicals show fluorescence with short-wavelength emission (444–529 nm). Theoretical calculations and experimental results suggest that the short-wavelength emission in the current radicals can be attributed to anti-Kasha's rule behavior, in which the emission originates from the relatively high excited states ($D_3/D_2$) to $D_0$ (Fig. 1b)[19–24]. This kind of radical is advantageous because they have a wide bandgap, and they exhibit blue-shifted absorption and emission. Thus, they are intrinsic candidates for potential applications for blue absorption and emission.

## Results

### Synthesis of the NHC-based precursors

NHCs featuring bulky substituents and an empty p orbital offer excellent platforms to stabilize radical species[25–33]. Moreover, the electronic

[1]Key Laboratory of Synthetic and Natural Functional Molecule of the Ministry of Education, College of Chemistry and Materials Science, Northwest University, Xi'an 710127, People's Republic of China. ✉e-mail: yfhan@nwu.edu.cn

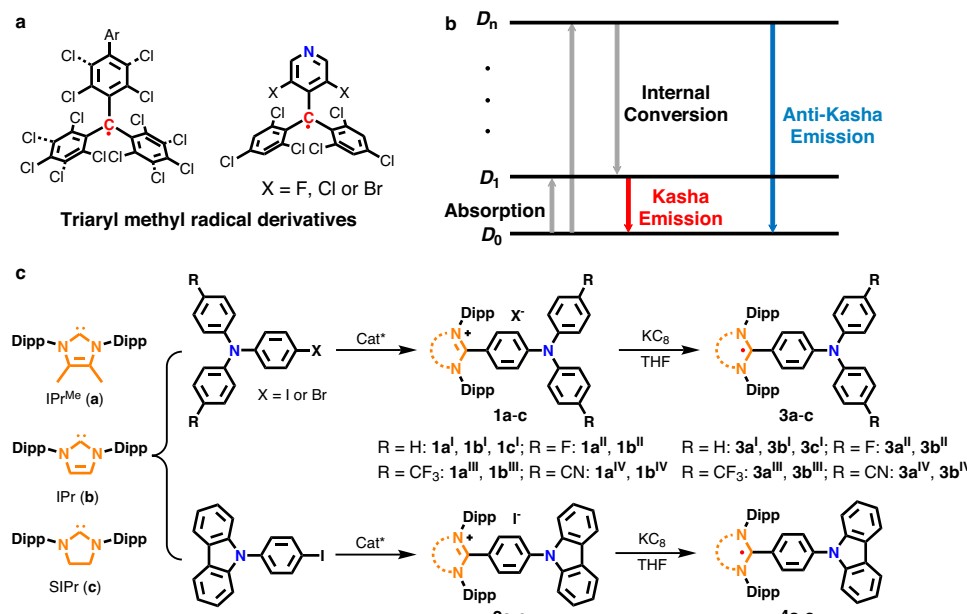

**Fig. 1 | Previous luminescent organic radical structural motifs, the emissive processes of relevance to this study, and the synthesis of NHC-based precursors 1a-c, 2a-c, and organic radicals 3a-c, 4a-c. a** Chemical structures of the reported triaryl methyl radical derivatives. So far, luminescent radicals are mainly confined in triaryl methyl radical derivative, the emission wavelengths are limited to red and NIR regions based on Kasha' rule. **b** The schematic illustration of the processes of Kasha and anti-Kasha emission. The Kasha's rule applies to most organic molecules. It states that the emitting electronic level of a given multiplicity is the lowest excited level of that multiplicity. Few organic compounds exhibit anomalous emission from relatively high excited states; they are exceptions to Kasha's rule. These anti-Kasha's rule emissions are mostly characteristic short-wavelength bands in emission spectra; they avoid additional consumption from internal conversions and other forms of electronic relaxation processes. **c** The synthetic pathway of the NHC-based radicals studied in this work. The desired precursor salts **1a–c** and **2a–c** are readily accessible by the direct C2-arylation of NHCs (**a–c**) with $Pd_2(dba)_3/Ni(COD)_2$ as a catalyst. The subsequent reduction of **1a–c** and **2a–c** with $KC_8$ affords radicals **3a–c** and **4a–c** in tetrahydrofuran (THF). Dipp, 2,6-diisopropylphenyl.

properties of the organic radicals can be effectively tuned by varying the carbene units[34–36]. The target molecules were chosen according to the ring backbone of the NHCs and the carbon center-attached units (Fig. 1c). The radical precursors (**1a–c** and **2a–c**) were readily synthesized by conducting the C2-arylation of the corresponding free NHCs (IPr$^{Me}$ = 4,5-dimethyl-N,N´-bis(2,6-diisopropylphenyl)imidazol-2-ylidene, **a**; IPr = 1,3-bis-(2,6-diisopropylphenyl)imidazol-2-ylidene, **b**; SIPr = 1,3-bis(2,6-diisopropylphenyl)imidazolidin-2-ylidene, **c**) with 4-iodo-N,N-diphenylaniline, as well as disubstituted TPA derivatives (−F, −CF₃, −CN) and 9-(4-iodophenyl)carbazole (CBP) in dioxane with $Pd_2(dba)_3/Ni(COD)_2$ as a catalyst (see Supplementary Pages 4–9 for more synthesis details)[37]. The precursor salts **1** and **2** were fully characterized by nuclear magnetic resonance (NMR) spectroscopy (Supplementary Figs. 1–24), high-resolution electrospray ionization (HR-ESI) mass spectrometry (Supplementary Figs. 25–36), and X-ray diffraction (XRD) studies (Figs. 2, 3, and Supplementary Figs. 37–41). The cyclic voltammograms of all the precursor salts showed a reversible redox-wave ($E_{1/2}$ = −2.07 to −1.61 V, against Fc$^+$/Fc) indicating the synthetic accessibility of the corresponding neutral radical species (Supplementary Figs. 42–44). The $E_{1/2}$ values were anodically shifted as the π-accepting properties of carbene ligands increased ($E_{1/2}$ = −2.07, −1.98, and −1.75 V for **1a$^I$**, **1b$^I$**, and **1c$^I$**; −1.92, −1.85, and −1.61 V for **2a**, **2b**, and **2c**, respectively). This shows that the electrochemical potential can be tuned by changing the carbene ligand. Moreover, the electron-withdrawing groups on the TPA increased the reduction potential of the $E_{1/2}$. Specifically, the $E_{1/2}$ values were anodically shifted by 150 mV for the CN-substitutional precursors **1a$^{IV}$** (−1.90 V) and **1b$^{IV}$** (−1.83 V) relative to the counterparts **1a$^I$** (−2.07 V) and **1b$^I$** (−1.98 V).

## Synthesis of the NHC-based radicals

The chemical reduction of precursor salts **1a–c** and **2a–c** with one equivalent $KC_8$ afforded neutral radicals **3a–c** and **4a–c** in good to excellent yields after workup. The paramagnetic character of **3a–c** and **4a–c** was first indicated by NMR spectra; all of them were NMR silent. The radical nature of **3a–c** and **4a–c** was further confirmed by X-band electron paramagnetic resonance, which was successfully simulated based on the calculated coupling constants (Supplementary Figs. 45–56). All radicals were stable in solution and in the solid state under an inert gas atmosphere. The TGA measurement shows that the 5% weight-loss temperature of all radicals is higher than 170 °C (Supplementary Figs. 57–60). In contrast to IPr-based radicals **3b$^I$** (171 °C) and **4b** (179 °C), SIPr-based radicals **3c$^I$** (230 °C) and **4c** (257 °C) have significantly higher thermal decomposition temperature. These results indicated that the saturated NHC (SIPr) is beneficial in increasing the thermal decomposition temperature of radicals. In addition, the photostability was investigated by measuring the decay of fluorescence intensity upon continuous photoirradiation (370 nm excitation wavelength) in THF. All radicals displayed excellent photostability in solution ($t_{1/2}$ > 1×10⁴ s, Supplementary Figs. 61–63). The multicycle CV curves (25 cycles) show that either the shape, peak position, or the onset does not change, implying the decent stability of radicals in the process of oxidation and reduction (Supplementary Figs. 64–67). Furthermore, the selected radicals (for **3a$^{II}$** and **3b$^{II}$**) dispersed in polymethyl methacrylate (PMMA) film also exhibited unique photostability (Supplementary Fig. 68).

## Solid-state structural characterization

The structures of the precursor salts **1a$^I$**, **1b$^I$**, **1c$^I$**, and **1b$^{III}$**; **2a–c** and the radicals **3a$^I$**, **3b$^I$**, **3c$^I$**, **3b$^{III}$**; and **4a–c** were determined by XRD. Each of the radicals showed that the carbene carbon atom (C1) was sp² hybridized. All the radicals showed shorter C1–C2 bond (1.397(3)–1.416(2) Å) length than the precursors (1.457(5)–1.472(3) Å); this suggests the presence of a C–C double bond character in all the radicals (Figs. 2 and 3). The C1–N1/N2 bond lengths of the TPA-radicals **3a$^I$** (av. 1.401 Å), **3b$^I$**

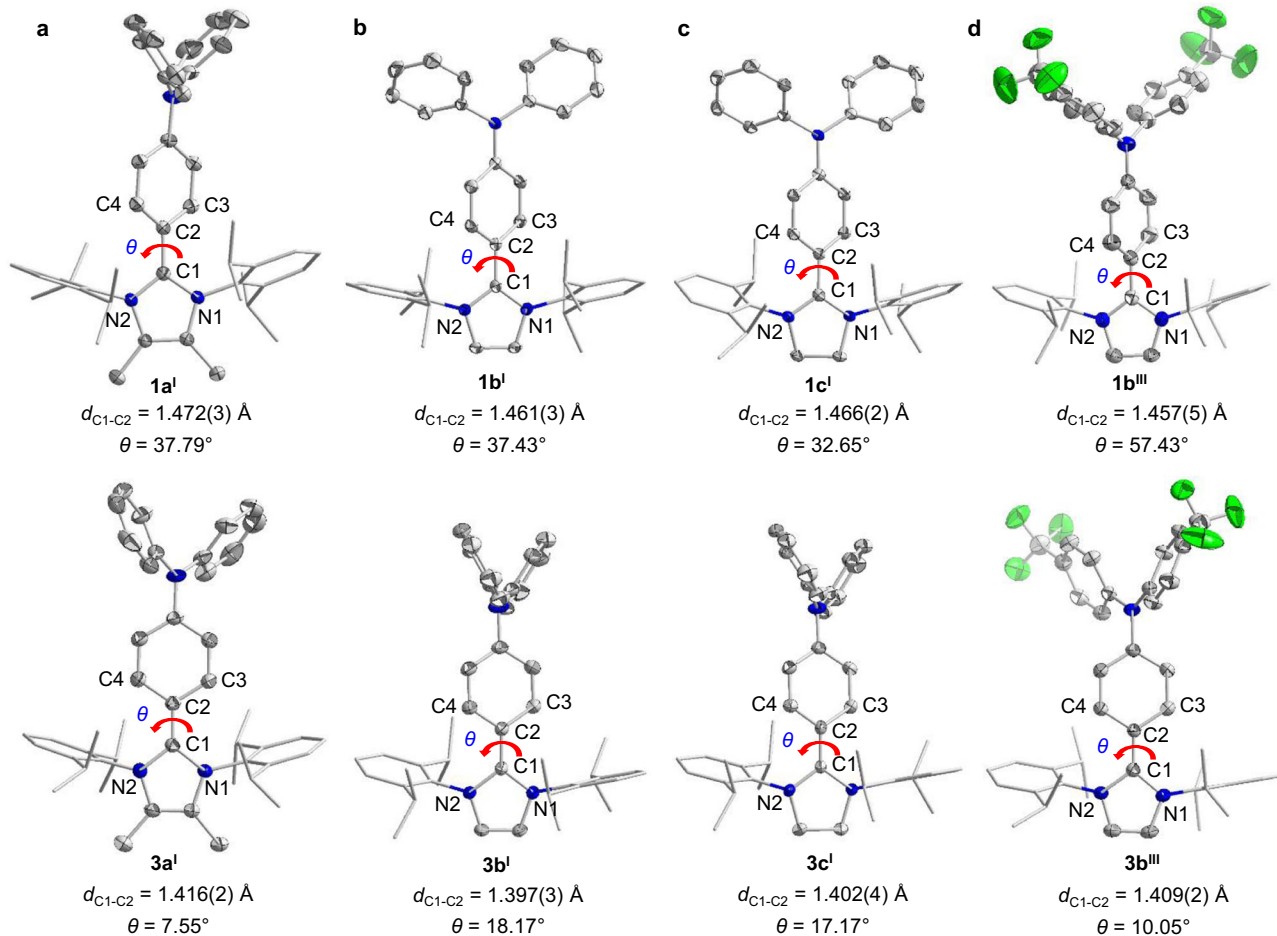

**Fig. 2 | X-ray solid-state structures of the TPA–NHC precursors and their corresponding carbon-centered radicals. a** X-ray solid-state structures for precursor **1a^I** and radical **3a^I**. **b** X-ray solid-state structures for precursor **1b^I** and radical **3b^I**. **c** X-ray solid-state structures for precursor **1c^I** and radical **3c^I**. **d** X-ray solid-state structures for precursor **1b^III** and radical **3b^III**. $\theta$: the twisted angle between the C3–C2–C4 and N1–C1–N2 plane. The single crystals of radical precursors **1a^I**, **1b^I**, **1c^I**, and **1b^III** were grown by slowly diffusing diethyl ether into a saturated acetonitrile solution at ambient temperature. The single crystals of radicals **3a^I**, **3b^I**, **3c^I**, and **3b^III** were obtained from a saturated hexane solution upon cooling to −30 °C. Thermal ellipsoids are shown with 50% probability. Color scheme: C, gray; N, blue; F, green. Hydrogen atoms, the counter anion, and solvent molecules are omitted for clarity. Selected bond lengths and angles Å and deg: (**1a^I**): C1–N1 1.348(3); C1–N2 1.351(3); <N1–C1–N2 106.47(17). (**1b^I**): C1–N1 1.349(3); C1–N2 1.353(3); <N1–C1–N2 106.65(19). (**1c^I**): C1–N1 1.334(2); C1–N2 1.337(2); <N1–C1–N2 110.84(13). (**1b^III**): C1–N1 1.343(4); C1–N2 1.345(4); <N1–C1–N2 106.9(3). (**3a^I**): C1–N1 1.399(2); C1–N2 1.403(2); <N1–C1–N2 103.81(14). (**3b^I**): C1–N1 1.398(2); C1–N2 1.402(2); <N1–C1–N2 105.04(14). (**3c^I**): C1–N1 1.397(4); C1–N2 1.400(4); <N1–C1–N2 108.5(3). (**3b^III**): C1–N1 1.4019(19); C1–N2 1.3974(18); <N1–C1–N2 104.33(12).

(av. 1.400 Å), and **3c^I** (av. 1.399 Å) were considerably longer than those of **1a^I** (av. 1.350 Å), **1b^I** (av. 1.351 Å), and **1c^I** (av. 1.336 Å), while the N1–C1–N2 bond angle of **3a^I**, **3b^I**, and **3c^I** was slightly smaller than that of **1a^I**, **1b^I**, and **1c^I**. Further, the similar phenomenon was observed in CBP-radicals **4a**, **4b**, and **4c**. Additionally, after the one-electron reduction of the precursors to radicals, the twisted angle between the N1–C1–N2 (carbene) plane and C3–C2–C4 (the central *p*-phenylene bridge) significantly decreased, particularly in radical **4a**: the twisted angle was only 4.73° (Fig. 3). Furthermore, the C–C bond length of the central *p*-phenylene bridge was observed. The changes in the structural parameter indicate the enhanced conjugation degree of the carbene ligand and the central *p*-phenylene bridge in the radicals and the partial delocalization of the unpaired electron over the aryl substituent at the carbene carbon atom. In addition, the carbazolylpheyl substituents are more likely to form non-covalent intermolecular interactions such as π-stacking and C–H/π forces than the corresponding triphenylamine groups (Supplementary Figs. 69–71).

## DFT calculations
The electronic structures of NHC-based radicals **3a**−**c** and **4a**−**c** were further investigated by DFT calculations at the UB3LYP/6-

31G(d) level of theory. The results indicate that the carbene ligands negligibly affect the electron density distributions of the frontier molecular orbitals in **3a**−**c** and **4a**−**c** (Supplementary Figs. 72–83). The electron cloud of the singly occupied molecular orbital (SOMO) and the lowest singly unoccupied molecular orbital (SUMO) in **3a**−**c** and **4a**−**c** was mainly distributed on the carbene carbon atom and extended partially to the para and ortho positions of the central *p*-phenylene bridge; this is consistent with the single crystal structure. However, the energy of the frontier molecule orbital (SOMO and SUMO) decreased as the π-accepting capability of the carbene ligands increased from IPr^Me to SIPr^38–40 (Fig. 4a). The energy of the frontier molecule orbital was further reduced by increasing the electron-withdrawing character of the substituent group on the TPA (Fig. 4a). The Mulliken spin densities on the carbene carbon atom gradually increased with the increase in the π-accepting capability of the carbene moiety, whereas it decreased with the increase in the electron-withdrawing capabilities of the substituent (Fig. 4b). These results suggest that the electronic structure of the NHC-based radicals can be easily regulated by changing the ring backbone of the NHCs or the substituent group on TPA.

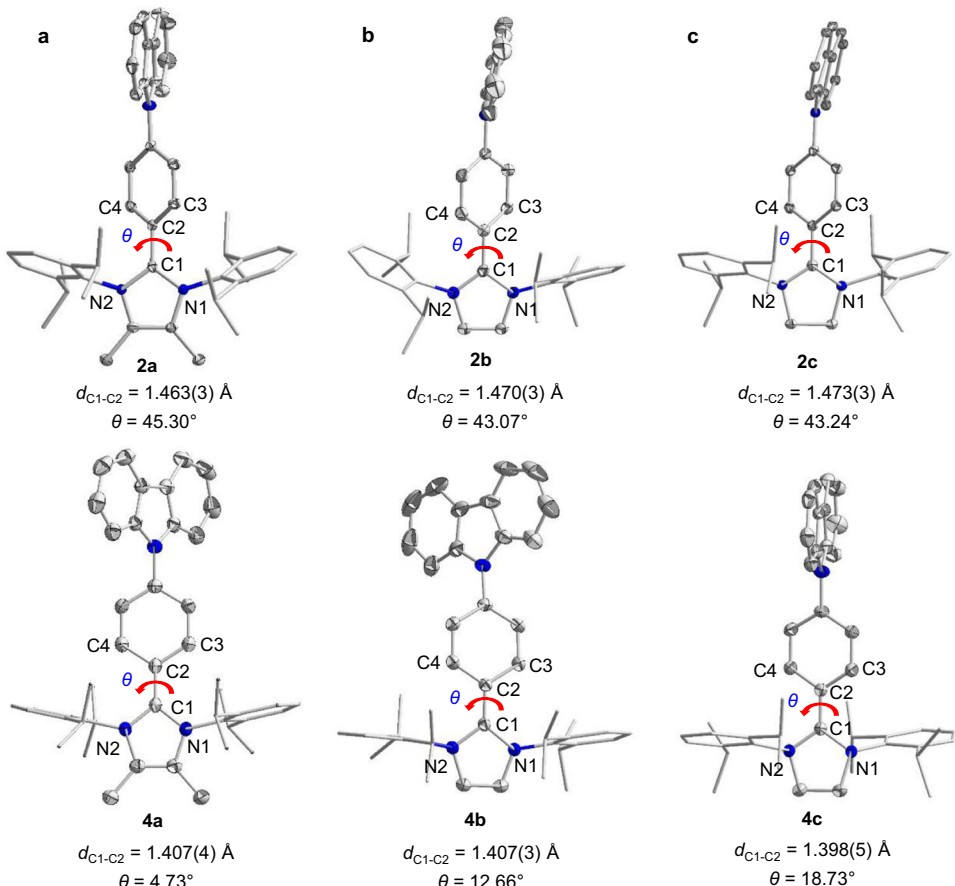

**Fig. 3 | X-ray solid-state structures of the CBP–NHC precursors and their corresponding carbon-centered radicals. a** X-ray solid-state structures for precursor **2a** and radical **4a**. **b** X-ray solid-state structures for precursor **2b** and radical **4b**. **c** X-ray solid-state structures for precursor **2c** and radical **4c**. $\theta$: the twisted angle between the C3–C2–C4 and N1–C1–N2 plane. The single crystals of radical precursors **2a**–**c** were grown by slowly diffusing diethyl ether into a saturated acetonitrile solution at ambient temperature. The single crystals of radicals **4a**–**c** can be obtained from a saturated hexane solution after cooling to −30 °C. Thermal ellipsoids are shown with 50% probability. Color scheme: C, gray; N, blue. Hydrogen atoms, the counter anion, and solvent molecules are omitted for clarity. Selected bond lengths and angles Å and deg: (**2a**): C1–N1 1.347(3); C1–N2 1.351(3); <N1–C1–N2 107.3(2). (**2b**): C1–N1 1.340(3); C1–N2 1.349(3); <N1–C1–N2 107.2(2). (**2c**): C1–N1 1.325(3); C1–N2 1.332(3); <N1–C1–N2 112.08(18). (**4a**): C1–N1 1.399(2); C1–N2 1.399(2); <N1–C1–N2 103.1(2). (**4b**): C1–N1 1.400(2); C1–N2 1.404(2); <N1–C1–N2 103.87(14). (**4c**): C1–N1 1.382(5); C1–N2 1.389(4); <N1–C1–N2 108.5(3).

## Photophysical properties

The ultraviolet–visible (UV–vis) absorption spectrum of radicals **3a**–**c** and **4a**–**c** recorded in THF was characterized by a new absorption band in the range of 460–680 nm compared to their corresponding precursors **1a**–**c** and **2a**–**c** (Supplementary Figs. 84–95). The TD–DFT calculations suggested that these absorptions were attributed to SOMO→LUMO+3 (for **3a**[I]–**3a**[IV], **4a**, **4b**), SOMO→LUMO+2 (for **3b**[I]–**3b**[IV], **3c**[I], **4c**), and SOMO→LUMO+4 (for **3b**[III]) transitions. The calculated lowest energy transition corresponded to the SOMO→LUMO transition with considerably low oscillator strength (*f*); therefore, the lowest energy transition did not contribute significantly to the absorption profile. The excitation of radicals **3a**–**c** and **4a**–**c** in THF solution at room temperature with visible light above 460 nm only produced weak photoluminescence, while the emission intensity was significantly enhanced when the excitation was performed with UV at 375 nm. However, radicals **3a**–**c** and **4a**–**c** was non-emissive in the solid states probably because of the aggregation-caused quenching, and this is consistent with previous literature results[9]. Figure 5a, b and Supplementary Figs. 96–105 show the emission spectra of radicals **3a**–**c** and **4a**–**c** in THF solution at room temperature, and their photophysical properties are summarized in Table 1. Although the absorption in the visible region is weak, all radicals show a negative Stokes shift suggesting that the emission doesn't occur from the lowest excited state. The emission color of NHC-based radicals **3a**–**c**

and **4a**–**c**, from blue to green (Fig. 5c), was significantly blue-shifted with respect to those of known triaryl methyl radical derivatives[8]. Furthermore, radicals **3a**[III], **3a**[IV], **3b**[III], and **3b**[IV] with strong electron-withdrawing groups (–CF₃ and –CN) on TPA had significantly shorter emission wavelength than their radical counterparts **3a**[I] and **3b**[I], while the emission wavelength of fluorine-substituted radical derivatives **3a**[II] and **3b**[II] remained nearly unchanged (Table 1). Additionally, the emission wavelength of CBP–NHC radicals **4a**–**c** was blue-shifted compared with that of their TPA–NHC radical counterparts **3a**[I]–**3c**[I]. Therefore, we can conclude that the emission wavelength of the NHC-based radicals blue shifted with the decrease in the SOMO energy. However, the introduction of carbazolylpheyl substituents in our radical system did not achieve the desired emission property compared to TPA-based species probably caused by the existence of multiple intermolecular interactions. All IPr^Me/IPr-based radicals showed absolute fluorescence quantum yields ($\Phi_f$) from 6.5 to 36.8%, which were superior to those of SIPr-based radicals **3c**[I] (1.5%) and **4c** (0.1%). Notably, the CIE (Commission Internationale de L'Eclairage) chromaticity coordinates of **3a**[I] (0.30, 0.44), **3b**[II] (0.25, 0.37), and **3a**[III] (0.17, 0.21) corresponded to green, cyan, and blue, respectively, with high quantum yields, making them promising as short-wavelength emitters in OLEDs (Supplementary Figs. 106–108). The short excited-state luminescence lifetime of radicals **3a**–**c** and **4a**–**c** suggests that the emission was fluorescent in nature (Table 1). Owing to the existence of an unpaired electron, the

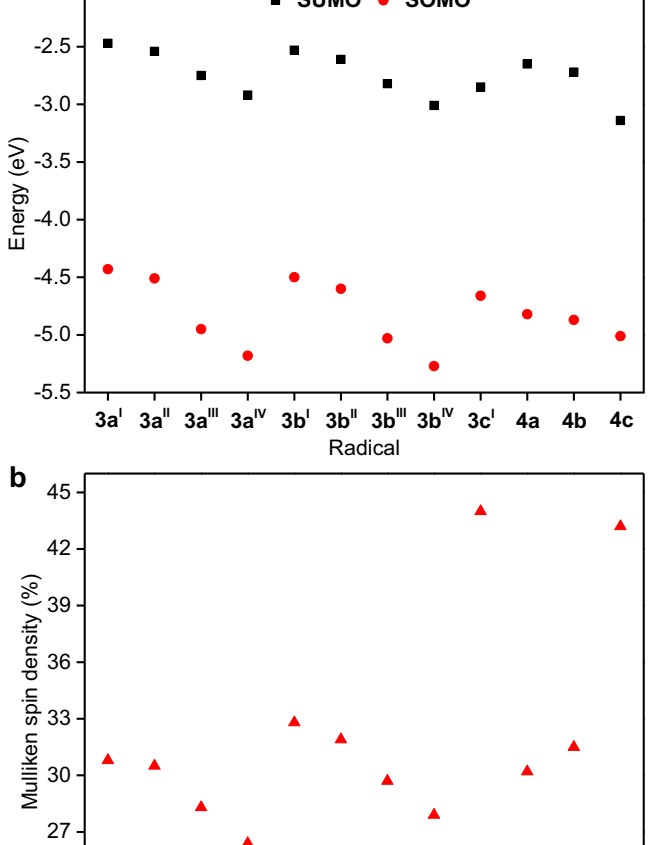

**Fig. 4 | Quantum-chemical results for radicals 3a–c and 4a–c. a** DFT energies of the SOMO and SUMO frontier orbitals of radicals **3a–c** and **4a–c**, as calculated at the tuned UB3LYP/6-31G(d) level of theory. SOMO, singly occupied molecular orbital; SUMO, lowest singly unoccupied molecular orbital. **b** Calculated Mulliken spin densities on carbene carbon atom in radicals **3a–c** and **4a–c**.

affect the energy levels of the SOMO and emission wavelength of the radicals. Compared to the SIPr-stabilized radical, the IPr and IPr$^{Me}$-stabilized radical exhibited a red-shifted emission and a relatively high quantum efficiency. Additionally, the strong electron-withdrawing substituents (–CF$_3$ and –CN) on TPA can further promote the blue shift of the emission spectrum. According to the experimental results and TD–DFT calculation, the short-wavelength emission of the NHC-based radicals originate from the relatively high excited states rather than the lowest excited state, and this violates Kasha's rule compared to those of known triaryl methyl radical derivatives. Notably, we can construct blue, cyan, and green luminescent radicals with high quantum efficiency (>20%) through the proper choice of carbene ligands and TPA units. Moreover, we illustrate that the anti-Kasha's rule is expected to guide the synthesis of new organic luminescent radicals with high-energy emission. This finding paves the way for the discovery of new organic radicals with rare blue fluorescence, which opens the possibility of the generation of radical-based blue-emitting OLEDs.

## Methods

Experiments were performed under an inert gas (Ar or N$_2$) atmosphere using standard Schlenk techniques and glovebox. THF, dioxane and hexane were dried by standard methods and distilled under nitrogen. $^1$H, $^{13}$C{$^1$H} were recorded on Bruker AVANCE III 400. Chemical shifts ($\delta$) are expressed in ppm downfield from tetramethylsilane using the residual protonated solvent as an internal standard. Mass spectra were obtained with a Bruker microTOF-Q II mass spectrometer (Bruker Daltonics Corp., USA) in the electrospray ionization (ESI) mode. IR spectra were recorded in Nujol oil using KBr plates as the infrared transmission window on a Nicolet AVATAR 360 FT-IR spectrometer. Elemental analyses were carried out on an Elementar VarioEL III instrument.

### EPR measurements

The continuous wave (CW) EPR spectra were obtained using an X-band Bruker E500 spectrometer at room temperature. Diffraction data of compound were collected with a Bruker APEX-II CCD diffractometer.

### CV measurement

CV experiments were carried out using a CHI660E electrochemical workstation (CH Instruments, Inc). All experiments were carried out under an atmosphere of nitrogen in degassed and anhydrous acetonitrile solution containing Bu$_4$NPF$_6$ (0.1 M) at a scan rate of 100 mV s$^{-1}$. The setup consisted of a glassy carbon working electrode, a platinum wire counter electrode, and a silver wire inserted in a small glass tube fitted with a porous Vycor frit and filled with a AgNO$_3$ solution in acetonitrile (0.01 M). Ferrocene was used as a standard, and all reduction potentials are reported with respect to the $E_{1/2}$ of the Fc$^+$/Fc redox couple.

### Spectral measurements

UV–visible spectra were recorded using an Agilent Technologies Evolution 300 spectrophotometer. The fluorescence experiments were performed on a Horiba Fluorolog-3 spectrometer. Fluorescence decay profiles were recorded on a FLS920 instrument. The experimental quantum yields were determined by recording the emission signals within an integrating light sphere on a FLS980 Photoluminescence Spectrometer (Edinburgh Instruments) equipped with an ozone-free Xenon Arc Lamp (450 W), photomultiplier R928P and double grating excitation and emission monochromators (Czerny-Turner type).

### Computational methodology

DFT calculations were executed using the Gaussian 09 program package (rev. E01)[44]. The geometries of the compounds were optimized without symmetry constraints using the crystal structure coordinate as the starting structure. Calculations were performed using the unrestricted Becke three-parameter hybrid functional with

ground states of **3a–c** and **4a–c** were doublet. The TD–DFT calculations (wB97X/6-31G(d,p))[41] suggested that the lowest excited state (D$_1$) can be mainly assigned to the transition from $\alpha$-LUMO to SOMO. However, the oscillator strength ($f$) of D$_1$ in these radicals was almost zero, which is indicative of the dark state nature of D$_1$[42,43]. The relatively high excited states (D$_2$ and D$_3$) of radicals **3a–c** and **4a–c** afforded high oscillator strength. In our current system (except for **3a$^{IV}$** and **3b$^{IV}$**), the energy gap between D$_2$ and D$_1$ is higher than 0.46 eV, which is three times that of known triaryl methyl radical derivatives[12]. It makes the internal conversion slow and favorable to the emission of anti-Kasha. Therefore, we infer that the efficient emission of **3a–c** and **4a–c** most likely originated from D$_2$ or D$_3$, which corresponds with a radiative relaxation of $\alpha$-LUMO+1 or $\alpha$-LUMO+2 to SOMO and does not obey Kasha's rule. Compared to the general Kasha's rule, the anti-Kasha's rule from relatively high excited states (D$_n$ → D$_0$) is beneficial for obtaining organic radicals with short-wavelength emission, as their energy gap is larger than that for D$_1$ → D$_0$[20, 21].

## Discussion

In conclusion, our NHC-based design offers a platform to seek a new class of organic radical materials that display tunable luminescence from blue to green. The choice of the ring backbone of the NHC ligands and the electronic properties of the substituents on TPA both strongly

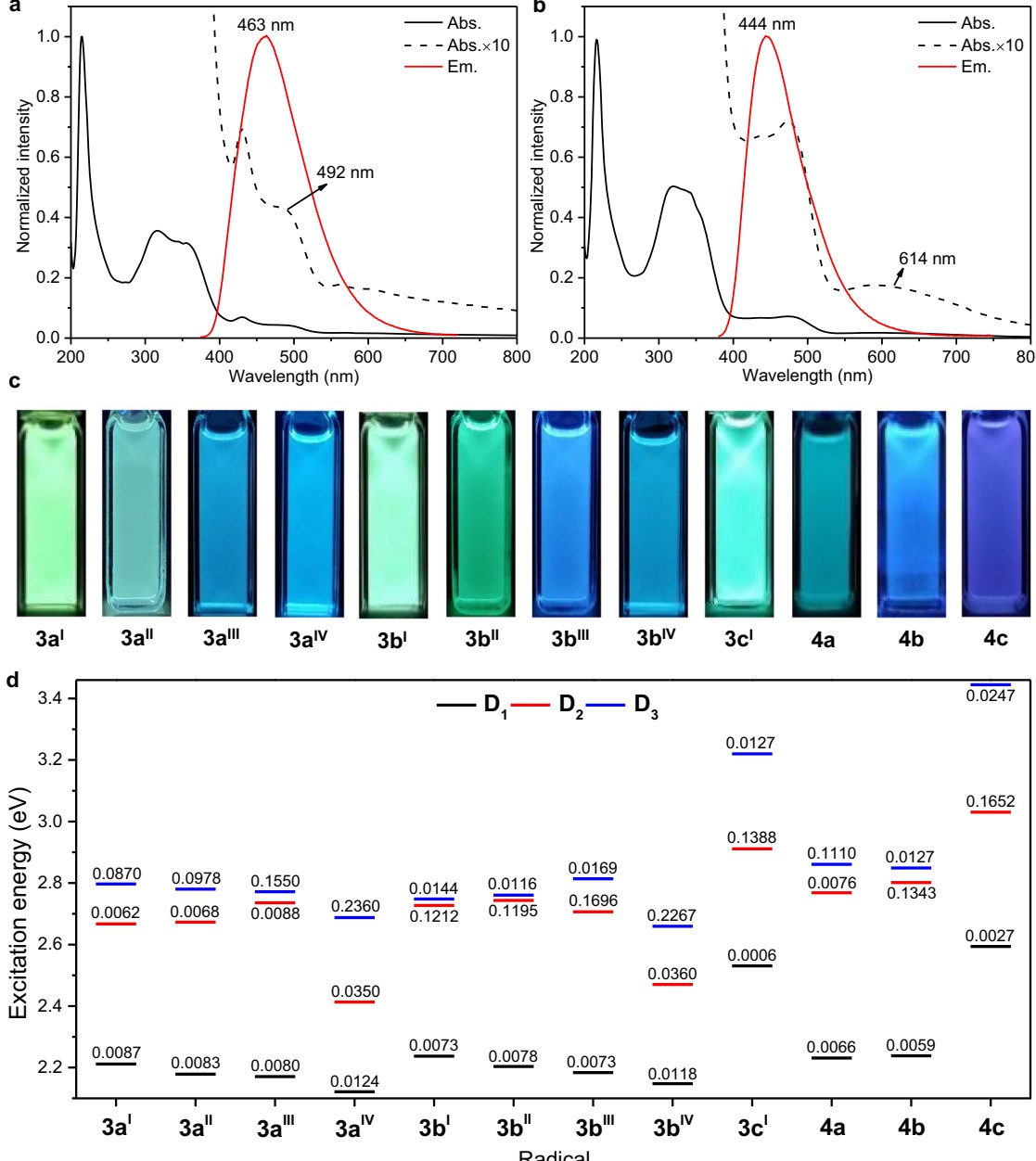

**Fig. 5 | Emission spectra and DFT energies of the doublet (D₁-D₃) states of the NHC–based radicals. a** Absorption and emission spectra of **3aᴵᴵᴵ** in THF at room temperature. Enlarged portion of absorption spectra (10 fold) are shown. **b** Absorption and emission spectra of **3bᴵᴵᴵ** in THF at room temperature. Enlarged portion of absorption spectra (10 fold) are shown. **c** Photographs of the as-synthesized radicals in THF under irradiation with UV light at $\lambda$ = 365 nm. **d** The DFT energies of the doublet (D₁–D₃) states and their corresponding oscillator strengths (*f*) of radicals **3a**–**c** and **4a**–**c**, as calculated by the wB97X/6-31G(d,p) level of theory with polarizable continuum model solvation in THF.

Lee−Yang−Parr correlation functional (B3LYP)[45] with the 6-31G(d) basis set. Frequency calculations were carried out to ensure that the optimized geometries were minima on the potential energy surface, in which no imaginary frequencies were observed in any of the compounds. The explicitly spin-adapted TDDFT (X-TDDFT) is performed to study the excited-state properties using the Beijing Density Functional (BDF) package (BDF-G)[46].

## TGA measurement
Thermal gravimetric analysis (TGA) was carried out on the Pyris1 TGA thermal analysis system at a heating rate of 10 °C min⁻¹ under nitrogen protection.

## Photostability measurements
In the argon glove box, the radicals were dissolved into THF with a concentration of $5 \times 10^{-5}$ mol L⁻¹ and sealed in 1 cm-optical-path-length quartz cells and then set at a Horiba Fluorolog-3 spectrometer. The intensity of the maximum emission peak was recorded exciting at 370 nm light. The logarithm of fluorescence intensity versus time was plotted, and a slope of the approximate line was estimated to be a photolysis rate.

## Redox stability measurement
All experiments were carried out under an atmosphere of argon in degassed and anhydrous acetonitrile solution containing *n*-Bu₄NPF₆

**Table 1 | Photophysical parameters of NHC-based luminescent radicals 3a–c and 4a–c in THF**

| Radical | $\lambda_{ex}$ (nm) | $\lambda_{em}$ (nm) | $\tau$ (ns) | $\Phi_f$ (%) |
|---------|------|------|------|------|
| 3a[I]   | 376  | 529  | 6.9  | 33.1 |
| 3a[II]  | 377  | 520  | 4.3  | 24.8 |
| 3a[III] | 364  | 463  | 4.4  | 32.6 |
| 3a[IV]  | 372  | 460  | 3.6  | 23.8 |
| 3b[I]   | 375  | 490  | 6.1  | 32.7 |
| 3b[II]  | 376  | 499  | 6.5  | 36.8 |
| 3b[III] | 370  | 444  | 3.5  | 10.4 |
| 3b[IV]  | 370  | 464  | 3.4  | 14.3 |
| 3c[I]   | 370  | 484  | 5.1  | 1.5  |
| 4a      | 365  | 482  | 8.1  | 6.5  |
| 4b      | 372  | 470  | 7.2  | 13.4 |
| 4c      | 376  | 469  | 5.8  | 0.1  |

(0.1 M) at a scan rate of $100 \, mV \, s^{-1}$. The setup consisted of a glassy carbon working electrode, a glassy carbon counter electrode, and a silver wire immersed in a saturated LiCl solution in EtOH and 0.1 M $n$-Bu$_4$NPF$_6$ solution in acetonitrile as the reference electrode. The recorded voltammograms were referenced to the internal standard Fc$^+$/Fc (ferrocenium/ferrocene) couple.

## Data availability

The authors declare that all data supporting the findings of this study are available within the article and Supplementary Information files, and are also available from the corresponding author upon request. The X-ray crystallographic coordinates for structures have been deposited at the Cambridge Crystallographic Data Center (CCDC) under deposition numbers CCDC-2107665 (**1a[I]**), CCDC-2160168 (**1a[II]**), CCDC-2160169 (**1a[III]**), CCDC-2160170 (**1a[IV]**), CCDC-2107667 (**1b[I]**), CCDC-2160163 (**1b[II]**), CCDC-2160165 (**1b[III]**), CCDC-2160166 (**1b[IV]**), CCDC-2107668 (**1c[I]**), CCDC-2107672 (**2a**), CCDC-2107673 (**2b**), CCDC-2107674 (**2c**), CCDC-2107669 (**3a[I]**), CCDC-2107670 (**3b[I]**), CCDC-2160172 (**3b[III]**), CCDC-2107671 (**3c[I]**), CCDC-2107675 (**4a**), CCDC-2107676 (**4b**), CCDC-2107677 (**4c**), respectively. These data can be obtained free of charge via http://www.ccdc.cam.ac.uk/data_request/cif.

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

## Acknowledgements

The authors gratefully acknowledge financial support from the National Natural Science Fund for Distinguished Young Scholars of China (No. 22025107, receipted by Y.-F.H.), the National Youth Topnotch Talent Support Program of China (Receipted by Y.-F.H.), Xi'an Key Laboratory of Functional Supramolecular Structure and Materials, and the FM&EM International Joint Laboratory of Northwest University.

## Author contributions

Y.-F.H. supervised the project. Y.-F.H. conceived and designed the experiments. X.L., Y.-L.W., C.C., and Y.-Y.R. performed the experiments. Y.-F.H., X.L., and Y.-L.W. co-wrote the paper. Y.-F.H., X.L., Y.-L.W., C.C., and Y.-Y.R. analyzed the data. All authors discussed the results in detail and commented on the manuscript.

## Competing interests

The authors declare no competing interests.
