## [Peer Review File · Nature Communications]

Reviewers' Comments:

Reviewer #1:

Remarks to the Author:

A Platform for Blue-Luminescent Carbon-Centered Radicals

This paper represents a step forward in the search for new organic radical materials emitting at low wavelengths (blue-green) thus, opening the possibility of generating radical-based blue-emitting OLEDs, which up to date are missing. Up to now, there are only deep-red and near-infrared radical OLEDs. Thus, it is a nice contribution of the presented research in order to overcome the efficiency limitation of OLEDs taking advantage of the unique doublet spin-configuration of the organic radicals and extending the emissions in the short-wavelength region. However, the stability of the reported radicals at ambient conditions (r.t., oxygen, photostability, etc) is not commented in the paper which is a key issue for such a kind of application.

More specifically, the paper describes the use of N-heterocyclic carbenes (NHCs) as a new kind of organic radical material which are interesting because they present short-wavelength luminescent. The authors have synthesized a family of carbon-centered N-carbene radicals incorporating different units like carbazolyphenyl (CBP) or diphenylaminophenyl (TPA) to the central NHC structure. Moreover, by adding substituents with different electron-withdrawing characters to the TPA unit, it has been demonstrated that it is possible to tune the luminescence further increasing the desired blue shift of the emission spectrum. A systematic and careful description of the synthesis and detailed characterization of all the synthesized compounds is presented in the supplementary information. It is worth noting that the authors have been able to crystallize all the synthesized compounds getting unequivocally information about the geometry of the molecules which is directly linked to the electronic structure and radical character of the new systems. Clear differences between the precursors (non-radical) and the radical structures are observed. Nevertheless, as already commented there is no study of the stability of the reported radicals under ambient conditions.

Reported theoretical calculations support the experimental findings demonstrating the short-wavelength emission comes from highly excited states rather than the lowest ones violating Kasha's rule compared to the known aryl methyl radical derivatives, the only radicals that emit up to now but only at high wavelengths. More specifically, the paper demonstrates that the anti-Kasha's rule from relatively high excited states is the key to obtaining organic radicals with short-wavelength emission, as their energy gap is larger compared with the general Kasha's rule described up to now. However, no discussion of the structural requirements to attain such an anti-Kasha rule is given which is also important to stress here.

In my opinion, the paper can be accepted for publication but only with major revisions which are mandatory. Important points to be included are:

1. Detailed experimental analysis of the thermal and photostability at ambient conditions of the new radicals should be added.
2. Rationalized why carbazolyphenyl substituents is not enhancing the desired emission property in comparison with the TPA.
3. The XY graphs for the chromaticity are nice to compare the shift of emissions but it'd be also nice to have some real pictures of the emission colors of the family of synthesized compounds.
4. Could the authors mention how the quenching of luminescence observed in solid-state will be surpassed in order to obtain emitting solid state devices like OLEDs?
5. The authors take advantage of the redox properties of the synthesized precursors and correlated the increase of electron withdrawing character of the different substituents with the reduction potential which decrease with increasing one of the substituents. Even though they are expected to be similar, it is surprising that the redox properties of the final radical species are not reported. Could the authors measure them?

Reviewer #2:

Remarks to the Author:

The authors address an important challenge of increasing the colour range for organic radicals that could be used for more efficient light-emitting diodes. In terms of significance, this is very important because current technologies based on triphenylmethyl radical generally have emission for orange, red, near-infrared. The work is certainly of interest to the community, but major revisions are recommended before consideration in Nature Communications.

To support claim of anti-kasha behaviour to allow new colour range:

- The experimental data (in SI) should be used to show the Anti-kasha emission claim in the main text. For example, plots of UV-vis absorption should be overlaid with emission using left and right y-axis. Figure 5 should be changed to include this, highlighting the negative Stokes shift.
- It is not clear that the source of higher energy emission ($> D1$) is not because of remaining precursor (e.g. 1a(i-iv) and 1b(i-iv)) in samples of 3a(i-iv) and 3b(i-iv). The authors should clarify this in the manuscript.
- The electronic structure calculations rely on UKS-TDDFT for excited states. It is very important that the authors clarify they have considered the spin contamination issue for reliability of excited state calculations. For example see: <https://pubs.acs.org/doi/10.1021/acs.jpcllett.8b03864>, which finds UKS-TDDFT reliable for D1 but more care is needed for higher energy states as proposed for the Anti-kasha emission. This is very important because it is part of the major claims of the paper, using theory to rationalize anti-kasha emission from relative oscillator strength of different excited states. It is important to clarify if the calculations are appropriate in not having a spin contamination problem for the radical excited states.
- The authors should clarify what drives emission from higher energy states (anti-kasha), and not internal conversion to a non-emissive D1. The excited state lifetimes of 3-8 ns in Table 1 should be considered with respect to competing internal conversion rates in the system, which presumably is due to energy gap law: to slow down internal conversion and favour higher excited state emission? This should be clarified in the manuscript.
- Following the previous point, I think the authors should give general design rules (e.g. some design limits such as energy difference between levels) so that the work can be translated to other structures, and therefore more applicable to a wider readership of the journal.

For significance to optoelectronics community:

- Stability of radicals is important for application in optoelectronics. The authors claim the materials are stable in solution and solid state, but no further details are given. The authors should give more information on at least the photostability and redox stability to assess practicality for light-emitting devices.
- All of the studies are done in solution. For relevance to optoelectronics as claimed here, the authors should try at least some characterisation of the behaviour in dilute films prepared by spin-coating or drop cast methods. For example a few % wt in PMMA, PVK or other wide bandgap host.
- Thermogravimetric analysis would be desirable to consider applicability of traditional vacuum deposition methods used in OLEDs for these materials.

Reviewer #3:

Remarks to the Author:

The present Manuscript reports on the synthesis, structural and photophysical characterization of a series of organic radicals obtained by incorporating either diphenylaminophenyl or carbazolylphenyl to N-heterocyclic carbenes. The Authors claim that the investigated radicals represent an exception to the Kasha rule, i.e. they emit from D2 or D3 doublet excited states rather than from D1. Such anomalous behavior explains their blue-shifted emission compared with other already known triaryl methyl radical derivatives.

Overall, the paper is certainly of interest for general readers involved in photophysics of organic molecular systems.

However, I have some concerns regarding the correct attribution of the observed emissions to anti-Kasha behavior. Looking at the computed levels reported as Supporting Information, the D2-D1 or D3-D1 energy gaps are not so high, and the oscillator strengths of the corresponding levels are not so different to justify anomalous emission of these compound. I think that additional evidence, possibly coming also from experiment, is imperative before claiming an anti-Kasha behavior.

- 1) How do the absorption spectra compare with excitation spectra?
- 2) How do the absorption spectra compare with the computed levels?
- 3) Since compounds are not emissive in solid state, measurements in blended films could provide additional information (e.g. emission from D1) on the emissive behavior of these compounds

Response to the Referees' Comments

Dear all reviewers,

We would like to sincerely thank you for the efficient work and constructive suggestions on our Manuscript "A Platform for Blue-Luminescent Carbon-Centered Radicals" (Manuscript ID: NCOMMS-22-15559-T). We are glad to see that all reviewers recommend its publication in *Nature Communications* and only some scientific revisions were suggested. Your comments are valuable and helpful for revising and improving our paper.

We have carefully revised the manuscript according to the suggestions of you. Some additional experiments were performed in response to the reviewers' comments and the corresponding results were provided. We have incorporated these changes (highlighted in yellow) into the edited version of the manuscript. Some changes in the Supporting Information were adjusted accordingly. The manuscript has been modified according to editorial requests. The detailed changes and corrections are listed in the response to the Referees' Comments (separately). Additional experiments were performed and some significant changes are listed below.

- 1) Detailed experimental analysis of the thermal and photostability at ambient conditions of the new radicals were provided. The redox property of the radical species was also further investigated and the result was provided.
- 2) We are very grateful to reviewer 2 for the suggestion on theoretical calculation and recommended relevant literature (<https://pubs.acs.org/doi/10.1021/acs.jpcllett.8b03864>). We were very happy to get in touch with the literature's corresponding author (Prof. Bingbing Suo) and get help in the theoretical calculation. The theoretical calculations of all radicals were further performed by using the spin-adapted time-dependent density functional theory (X-TDDFT), and the result indicated that the excited states (D_1 - D_6) in each radical are almost free from spin-contamination.
- 3) Based on additional experimental results and theoretical calculations, the anti-Kasha emission of as-synthesized radicals is further clarified. To highlight the negative Stokes shift in anti-Kasha emission, the real pictures of the emission colors of the radicals have been added to the revised manuscript.
- 4) The more details and discussions about the absorption and excitation spectra were added.
- 5) To illustrate the potential application of as-synthesized radicals in OLED, the fluorescence properties of the thin film of radicals in PMMA were preliminary studied.

We hope that after the positive reviews and addressing the points raised by the reviewers, the manuscript is now acceptable for publication in *Nature Communications*. Thanks again for your nice comments.

With best wishes

Authors

Reply to the comments by Referee 1

Reviewer #1 (Remarks to the Author):

A Platform for Blue-Luminescent Carbon-Centered Radicals

This paper represents a step forward in the search for new organic radical materials emitting at low wavelengths (blue-green) thus, opening the possibility of generating radical-based blue-emitting OLEDs, which up to date are missing. Up to now, there are only deep-red and near-infrared radical OLEDs. Thus, it is a nice contribution of the presented research in order to overcome the efficiency limitation of OLEDs taking advantage of the unique doublet spin-configuration of the organic radicals and extending the emissions in the short-wavelength region. However, the stability of the reported radicals at ambient conditions (r.t., oxygen, photostability, etc) is not commented in the paper which is a key issue for such a kind of application.

More specifically, the paper describes the use of N-heterocyclic carbenes (NHCs) as a new kind of organic radical material which are interesting because they present short-wavelength luminescent. The authors have synthesized a family of carbon-centered N-carbene radicals incorporating different units like carbazolylphenyl (CBP) or diphenylaminophenyl (TPA) to the central NHC structure. Moreover, by adding substituents with different electron-withdrawing characters to the TPA unit, it has been demonstrated that it is possible to tune the luminescence further increasing the desired blue shift of the emission spectrum. A systematic and careful description of the synthesis and detailed characterization of all the synthesized compounds is presented in the supplementary information. It is worth noting that the authors have been able to crystallize all the synthesized compounds getting unequivocally information about the geometry of the molecules which is directly linked to the electronic structure and radical character of the new systems. Clear differences between the precursors (non-radical) and the radical structures are observed. Nevertheless, as already commented there is no study of the stability of the reported radicals under ambient conditions.

Reported theoretical calculations support the experimental findings demonstrating the short-wavelength emission comes from highly excited states rather than the lowest ones violating Kasha's rule compared to the known aryl methyl radical derivatives, the only radicals that emit up to now but only at high wavelengths. More specifically, the paper demonstrates that the anti-Kasha's rule from relatively high excited states is the key to obtaining organic radicals with short-wavelength emission, as their energy gap is larger compared with the general Kasha's rule described up to now. However, no discussion of the structural requirements to attain such an anti-Kasha rule is given which is also important to stress here.

In my opinion, the paper can be accepted for publication but only with major revisions which are mandatory. Important points to be included are:

1. Detailed experimental analysis of the thermal and photostability at ambient conditions of the new radicals should be added.

Answer: Thank you for your comment. Detailed experimental conditions and experimental analysis of thermal and photostability of radicals have been added (please see page 3 in the revised main text). The TGA measurement shows that the 5% weight-loss temperature of all radicals is higher than 170 °C (Figures A1–A4). In contrast to IPr-based radicals **3b**¹ (171 °C) and **4b** (179 °C), SIPr-based radicals **3c**¹ (230 °C) and **4c** (257 °C) have significantly higher thermal decomposition temperature. These results indicated that the saturated NHC (SIPr) is beneficial in increasing the thermal decomposition temperature of radicals. In addition, the photostability was investigated by measuring the decay of fluorescence intensity upon continuous photoirradiation (370 nm excitation

wavelength) in THF (Figures A5–A8). All radicals displayed excellent photostability ($t_{1/2} > 1 \times 10^4$ s). For details, please see Figures S57–S64 in the Supporting Information. IPr = 1,3-bis-(2,6-diisopropylphenyl)imidazol-2-ylidene, SIPr = 1,3-bis-(2,6-diisopropylphenyl)imidazolidin-2-ylidene.

The details of experimental conditions are as follows:

TGA measurement. Thermal gravimetric analysis (TGA) was carried out on the Pyris1 TGA thermal analysis system at a heating rate of $10 \text{ }^\circ\text{C min}^{-1}$ under nitrogen protection.

Photostability measurements. In an argon glove box, the radicals were dissolved into THF with a concentration of $5 \times 10^{-5} \text{ mol L}^{-1}$ and sealed in 1-cm-optical-path-length quartz cells and then set at a Horiba Fluorolog-3 spectrometer. The intensity of the maximum emission peak was recorded exciting at 370 nm light. The logarithm of fluorescence intensity versus time was plotted, and a slope of the approximate line was estimated to be a photolysis rate.

Figure A1. TGA curve of radicals $3a^I$ – $3a^{IV}$ under nitrogen flow.

Figure A2. TGA curve of radicals $3b^I$ – $3b^{IV}$ under nitrogen flow.

Figure A3. TGA curve of radical $3c^I$ under nitrogen flow.

Figure A4. TGA curve of radicals 4a–c under nitrogen flow.

Figure A5. Plots showing the emission decay of radicals 3a^I–3a^{IV} in THF under continuous excitation with light at $\lambda = 370$ nm.

Figure A6. Plots showing the emission decay of radicals $3b^I$ – $3b^{IV}$ in THF under continuous excitation with light at $\lambda = 370$ nm.

Figure A7. Plots showing the emission decay of radical $3c^I$ in THF under continuous excitation with light at $\lambda = 370$ nm.

Figure A8. Plots showing the emission decay of radicals **3c^I** and **4a–c** in THF under continuous excitation with light at $\lambda = 370$ nm.

2. Rationalized why carbazolyphenyl substituents are not enhancing the desired emission property in comparison with the TPA.

Answer: Thank you for your comment. In general, increasing the rigidity of the whole molecule improves quantum yield by reducing thermal vibration and rotation of the molecules. The introduction of carbazolyphenyl substituents in our radical system did not achieve the desired emission property compared to TPA-based species probably caused by the existence of multiple intermolecular interactions. The carbazolyphenyl substituents are more likely to form non-covalent intermolecular interactions such as π -stacking and C-H/ π forces than the corresponding triphenylamine groups (Figures A9–A10). For details, please see Figures S69–S71 in the Supporting Information.

Figure A9. π -stacking interactions in carbazolyphenyl-based radical **4b**.

Figure A10. C-H/ π interactions in carbazolylpheyyl-based radical **4b**.

Figure A11. C-H/ π interactions in carbazolylpheyyl-based radical **4c**.

3. The XY graphs for the chromaticity are nice to compare the shift of emissions but it'd be also nice to have some real pictures of the emission colors of the family of synthesized compounds.

Answer: Thank you for your comment. Photographs of the as-synthesized radicals in THF under irradiation with UV light at $\lambda = 365$ nm were provided, please see Figure 5c in the revised main text. The XY graphs for the chromaticity have been transferred to the Supporting Information, please see Figures S106 and S107 in the Supporting Information.

Figure A12. Photographs of radical solutions in THF under irradiation with UV light at 365 nm.

4. Could the authors mention how the quenching of luminescence observed in solid-state will be surpassed in order to obtain emitting solid state devices like OLEDs?

Answer: Thank you for your suggestion. The aggregation-induced quenching (ACQ) in the condensed phase is a widespread phenomenon in many luminescent organic radical molecules. An effective way to reduce ACQ is to disperse luminescent organic radical molecules into host matrices. For example, efficient OLED devices by doping TTM/PTM-radical derivatives into matrix film have been described (*Nature*, **2018**, *563*, 536; *Nat. Mater.* **2020**, *19*, 1224). Our preliminary studies revealed that the doping of radicals into a PMMA matrix produces a strong fluorescence emission in the short-wave range, as observed in their THF solution (Figure A13).

Figure A13. The emission spectra of **3a^I** (left) and **3b^I** (right) with 5% wt in PMMA at room temperature recorded with a 370 nm excitation wavelength.

5. The authors take advantage of the redox properties of the synthesized precursors and correlated the increase of electron withdrawing character of the different substituents with the reduction potential which decrease with increasing one of the substituents. Even though they are expected to be similar, it is surprising that the redox properties of the final radical species are not reported. Could the authors measure them?

Answer: We thank the suggestions of the reviewer. The redox properties of the final radical species were investigated by using an electrochemical analyzer (CHI660E, CH Instruments). Cyclic voltammograms of all radicals showed a reversible redox wave at -2.07 to -1.61 V against Fc^+/Fc . Similar to the precursors, increasing the electron-withdrawing character of the substituent and the π -accepting capability of the carbene moiety will cause the $E_{1/2}$ values to be anodically shifted (Figures A14–A17).

The details of experimental conditions are as follows:

CV measurement. All experiments were carried out under an atmosphere of argon in degassed and anhydrous acetonitrile solution containing $n\text{-Bu}_4\text{NPF}_6$ (0.1 M) at a scan rate of 100 mV s^{-1} . The setup consisted of a glassy carbon working electrode, a glassy carbon counter electrode, and a silver wire immersed in a saturated LiCl solution in EtOH and 0.1 M $n\text{-Bu}_4\text{NPF}_6$ solution in acetonitrile as the reference electrode. The recorded voltammograms were referenced to the internal standard Fc^+/Fc (ferrocenium/ferrocene) couple.

Figure A14. Cyclic voltammograms of radicals $3a^I$ – $3a^{IV}$ in CH_3CN with 0.1 M Bu_4NPF_6 at a scan rate of 100 mV s^{-1} .

Figure A15. Cyclic voltammograms of radicals $3b^I$ – $3b^{IV}$ in CH_3CN with 0.1 M Bu_4NPF_6 at a scan rate of 100 mV s^{-1} .

Figure A16. Cyclic voltammograms of radical $3c^I$ in CH_3CN with 0.1 M Bu_4NPF_6 at a scan rate of 100 mV s^{-1} .

Figure A17. Cyclic voltammograms of radicals **4a–c** in CH_3CN with 0.1 M Bu_4NPF_6 at a scan rate of 100 mV s^{-1} .

Reply to the comments by Referee 2

Reviewer #2 (Remarks to the Author):

The authors address an important challenge of increasing the colour range for organic radicals that could be used for more efficient light-emitting diodes. In terms of significance, this is very important because current technologies based on triphenylmethyl radical generally have emission for orange, red, near-infrared. The work is certainly of interest to the community, but major revisions are recommended before consideration in Nature Communications.

To support claim of anti-kasha behaviour to allow new colour range:

- The experimental data (in SI) should be used to show the Anti-kasha emission claim in the main text. For example, plots of UV-vis absorption should be overlaid with emission using left and right y-axis. Figure 5 should be changed to include this, highlighting the negative Stokes shift.

Answer: We thank the suggestions of the reviewer. The UV-vis absorption and emission spectra have been modified, please see Figures 5a and 5b in the revised main text and Figures S96–S105 in the Supporting Information. Although the absorption of radicals in the visible region is weak, the characteristic negative Stokes shift can be observed (Figures B1–B12).

Figure B1. Absorption and emission spectra of $3a^I$ in THF at room temperature. Enlarged portion of absorption spectra (10 fold) are shown.

Figure B2. Absorption and emission spectra of $3a^{II}$ in THF at room temperature. Enlarged portion of absorption spectra (10 fold) are shown.

Figure B3. Absorption and emission spectra of $3a^{III}$ in THF at room temperature. Enlarged portion of absorption spectra (10 fold) are shown.

Figure B4. Absorption and emission spectra of $3a^{IV}$ in THF at room temperature. Enlarged portion of absorption spectra (10 fold) are shown.

Figure B5. Absorption and emission spectra of $3b^I$ in THF at room temperature. Enlarged portion of absorption spectra (10 fold) are shown.

Figure B6. Absorption and emission spectra of $3b^{II}$ in THF at room temperature. Enlarged portion of absorption spectra (10 fold) are shown.

Figure B7. Absorption and emission spectra of $3b^{III}$ in THF at room temperature. Enlarged portion of absorption spectra (10 fold) are shown.

Figure B8. Absorption and emission spectra of $3b^{IV}$ in THF at room temperature. Enlarged portion of absorption spectra (10 fold) are shown.

Figure B9. Absorption and emission spectra of **3c^I** in THF at room temperature. Enlarged portion of absorption spectra (10 fold) are shown.

Figure B10. Absorption and emission spectra of **4a** in THF at room temperature. Enlarged portion of absorption spectra (10 fold) are shown.

Figure B11. Absorption and emission spectra of **4b** in THF at room temperature. Enlarged portion of absorption spectra (10 fold) are shown.

Figure B12. Absorption and emission spectra of **4c** in THF at room temperature. Enlarged portion of absorption spectra (10 fold) are shown.

- It is not clear that the source of higher energy emission ($> D1$) is not because of remaining precursor (e.g. **1a(i-iv)** and **1b(i-iv)**) in samples of **3a(i-iv)** and **3b(i-iv)**. The authors should clarify this in the manuscript.

Answer: Thank you for your comment. We carefully re-checked the results and the experiments were investigated again. No residue of the precursor was observed. In addition, we also found that adding a small amount of precursor to the radical solution resulted in an obviously decrease in fluorescence intensity.

- The electronic structure calculations rely on UKS-TDDFT for excited states. It is very important that the authors clarify they have considered the spin contamination issue for reliability of excited state calculations. For example see: <https://pubs.acs.org/doi/10.1021/acs.jpcclett.8b03864>, which finds UKS-TDDFT reliable for $D1$ but more care is needed for higher energy states as proposed for the Anti-kasha emission. This is very important because it is part of the major claims of the paper, using theory to rationalize anti-kasha emission from relative oscillator strength of different excited states. It is important to clarify if the calculations are appropriate in not having a spin contamination problem for the radical excited states.

Answer: We are very grateful for your suggestion on theoretical calculation and recommended relevant literature (<https://pubs.acs.org/doi/10.1021/acs.jpcclett.8b03864>). We were very happy to get in touch with the literature's corresponding author (Prof. Bingbing Suo) and get help in the theoretical calculation. The theoretical calculations of all radicals were further performed by using the spin-adapted time-dependent density functional theory (X-TDDFT), and the result indicated that the excited states (D_1 - D_6) in each radical are almost free from spin-contamination.

- The authors should clarify what drives emission from higher energy states (anti-kasha), and not internal conversion to a non-emissive $D1$. The excited state lifetimes of 3-8 ns in Table 1 should be considered with respect to competing internal conversion rates in the system, which presumably is due to energy gap law: to slow down internal conversion and favour higher excited state emission? This should be clarified in the manuscript.

Answer: Thank you for your comment. According to the energy gap law, a large energy gap between

D₂ and D₁ is favorable to the emission from the highly excited state and unfavorable to the internal conversion process (*J. Am. Chem. Soc.* **2020**, *142*, 38; *Chin. J. Chem.* **2021**, *39*, 1297). In our current system (except for **3a^{IV}** and **3b^{IV}**), the energy gap between D₂ and D₁ is higher than 0.46 eV, which is three times that of known triaryl methyl radical derivatives (for example, *Angew. Chem. Int. Ed.* **2014**, *53*, 11845; *Angew. Chem. Int. Ed.* **2018**, *57*, 2869; *Phys. Chem. Chem. Phys.* **2018**, *20*, 18657). It makes the internal conversion slow and favorable to the emission of anti-Kasha. Relevant discussion has been added, please see page 7 in the revised main text.

- Following the previous point, I think the authors should give general design rules (e.g. some design limits such as energy difference between levels) so that the work can be translated to other structures, and therefore more applicable to a wider readership of the journal.

Answer: Thank you for your comment. The discussion about the design rule was added. The experimental results showed that the ring backbone of the NHCs had a significant effect on the emission behavior of radicals in current system. Specifically, we found that IPr^{Me}, IPr and SIPr-based radicals show anti-Kasha emission behavior while CAAC-based radicals exhibit Kasha emission, which may be due to the different π -accepting properties of NHC ligands. The investigation on mechanism is in progress. IPr^{Me} = 4,5-dimethyl-N,N'-bis(2,6-diisopropylphenyl)imidazol-2-ylidene, IPr = 1,3-bis-(2,6-diisopropylphenyl)imidazol-2-ylidene, SIPr = 1,3-bis-(2,6-diisopropylphenyl)imidazolidin-2-ylidene, CAAC = cyclic (alkyl)(amino)carbene.

For significance to optoelectronics community:

- Stability of radicals is important for application in optoelectronics. The authors claim the materials are stable in solution and solid state, but no further details are given. The authors should give more information on at least the photostability and redox stability to assess practicality for light-emitting devices.

Answer: We thank the suggestions of the reviewer. Detailed experimental conditions and experimental analysis of photostability and redox stability of radicals have been added. The photostability was investigated by measuring the decay of fluorescence intensity upon continuous photoirradiation (370 nm excitation wavelength) in THF (Figures B13–B16). All radicals displayed good photostability ($t_{1/2} > 1 \times 10^4$ s). In addition, the multicycle CV curves (25 cycles) show that either the shape, peak position, or the onset does not change, implying the decent stability of radicals in the process of oxidation and reduction (Figures B17–B20). For the description about photostability and redox stability of radicals, please see on page page 3 in the revised main text. For details, please see Figures S61–S68 in the Supporting Information.

The details of experimental conditions are as follows:

Photostability measurements. In an argon glove box, the radicals were dissolved into THF with a concentration of 5×10^{-5} mol L⁻¹ and sealed in 1-cm-optical-path-length quartz cells and then set at a Horiba Fluorolog-3 spectrometer. The intensity of the maximum emission peak was recorded exciting at 370 nm light. The logarithm of fluorescence intensity versus time was plotted, and a slope of the approximate line was estimated to be a photolysis rate.

Redox stability measurement. All experiments were carried out under an atmosphere of argon in degassed and anhydrous acetonitrile solution containing *n*-Bu₄NPF₆ (0.1 M) at a scan rate of 100 mV s⁻¹. The setup consisted of a glassy carbon working electrode, a glassy carbon counter electrode,

and a silver wire immersed in a saturated LiCl solution in EtOH and 0.1 M *n*-Bu₄NPF₆ solution in acetonitrile as the reference electrode. The recorded voltammograms were referenced to the internal standard Fc⁺/Fc (ferrocenium/ferrocene) couple.

Figure B13. Plots showing the emission decay of radicals $3a^I$ – $3a^{IV}$ in THF under continuous excitation with light at $\lambda = 370$ nm.

Figure B14. Plots showing the emission decay of radicals $\mathbf{3b}^{\text{I}}\text{--}\mathbf{3b}^{\text{IV}}$ in THF under continuous excitation with light at $\lambda = 370$ nm.

Figure B15. Plots showing the emission decay of radical $\mathbf{3c}^{\text{I}}$ in THF under continuous excitation with light at $\lambda = 370$ nm.

Figure B16. Plots showing the emission decay of radicals **4a–c** in THF under continuous excitation with light at $\lambda = 370$ nm.

Figure B17. The multicycle CV curves of radicals **3a^I–3a^{IV}** with scan rate of 100 mV s^{-1} .

Figure B18. The multicycle CV curves of radicals $3b^I$ – $3b^{IV}$ with scan rate of 100 mV s^{-1} .

Figure B19. The multicycle CV curves of radicals $3c^I$ with scan rate of 100 mV s^{-1} .

Figure B20. The multicycle CV curves of radicals **4a–c** with scan rate of 100 mV s^{-1} .

- All of the studies are done in solution. For relevance to optoelectronics as claimed here, the authors should try at least some characterisation of the behaviour in dilute films prepared by spin-coating or drop cast methods. For example a few % wt in PMMA, PVK or other wide bandgap host.

Answer: Thank you for your comment. According to the your suggestion, thin films of radicals in PMMA were prepared by drop-casting in THF solutions. Our preliminary studies revealed that the doping of radicals into a PMMA matrix produces a strong fluorescence emission in the short-wave range, as observed in their THF solution (Figure B21).

Figure B21. The emission spectra of **3a^I** (left) and **3b^I** (right) with 5% wt in PMMA at room temperature recorded with a 370 nm excitation wavelength.

- Thermogravimetric analysis would be desirable to consider applicability of traditional vacuum

deposition methods used in OLEDs for these materials.

Answer: Thank you for your comment. The TGA measurement shows that the 5% weight-loss temperature of all radicals is higher than 170 °C (Figures B22–B25). In contrast to IPr-based radicals **3b^I** (171 °C) and **4b** (179 °C), SIPr-based radicals **3c^I** (230 °C) and **4c** (257 °C) have significantly higher thermal decomposition temperature. These results indicated that the saturated NHC (SIPr) is beneficial in increasing the thermal decomposition temperature of radicals. We have added the description about the thermostability of radicals, please see page 3 in the revised main text. For details, please see Figures S57–S60 in the Supporting Information. IPr = 1,3-bis-(2,6-diisopropylphenyl)imidazol-2-ylidene, SIPr = 1,3-bis-(2,6-diisopropylphenyl)imidazolidin-2-ylidene.

The details of experimental conditions are as follows:

TGA measurement. Thermal gravimetric analysis (TGA) was carried out on the Pyris1 TGA thermal analysis system at a heating rate of 10 °C min⁻¹ under nitrogen protection.

Figure B22. TGA curve of radicals **3a^I**–**3a^{IV}** under nitrogen flow.

Figure B23. TGA curve of radicals $3b^I$ – $3b^{IV}$ under nitrogen flow.

Figure B24. TGA curve of radical $3c^I$ under nitrogen flow.

Figure B25. TGA curve of radicals 4a–c under nitrogen flow.

Reply to the comments by Referee 3

Reviewer #3 (Remarks to the Author):

The present Manuscript reports on the synthesis, structural and photophysical characterization of a series of organic radicals obtained by incorporating either diphenylaminophenyl or carbazolyphenyl to N-heterocyclic carbenes. The Authors claim that the investigated radicals represent an exception to the Kasha rule, i.e. they emit from D2 or D3 doublet excited states rather than from D1. Such anomalous behavior explains their blue-shifted emission compared with other already known triaryl methyl radical derivatives.

Overall, the paper is certainly of interest for general readers involved in photophysics of organic molecular systems.

However, I have some concerns regarding the correct attribution of the observed emissions to anti-Kasha behavior. Looking at the computed levels reported as Supporting Information, the D2-D1 or D3-D1 energy gaps are not so high, and the oscillator strengths of the corresponding levels are not so different to justify anomalous emission of these compound. I think that additional evidence, possibly coming also from experiment, is imperative before claiming an anti-Kasha behavior.

1) How do the absorption spectra compare with excitation spectra?

Answer: Thank you for your comment. The UV-vis absorption and excitation spectra have been modified, please see Figures S109–S117 in the Supporting Information. These results further confirm that the emission of radicals comes from a higher energy level than the lowest excited state (Figures C1–C9).

Figure C1. The absorption and excitation spectra of $3a^I$ ($\lambda_{em} = 529$ nm) in THF at room temperature.

Figure C2. The absorption and excitation spectra of **3a^{II}** ($\lambda_{em} = 520$ nm) in THF at room temperature.

Figure C3. The absorption and excitation spectra of **3a^{III}** ($\lambda_{em} = 463$ nm) in THF at room temperature.

Figure C4. The absorption and excitation spectra of **3a^{IV}** ($\lambda_{em} = 460$ nm) in THF at room temperature.

Figure C5. The absorption and excitation spectra of **3b^I** ($\lambda_{em} = 490$ nm) in THF at room temperature.

Figure C6. The absorption and excitation spectra of **3b^{II}** ($\lambda_{em} = 499$ nm) in THF at room temperature.

Figure C7. The absorption and excitation spectra of **3b^{III}** ($\lambda_{em} = 444$ nm) in THF at room temperature.

Figure C8. The absorption and excitation spectra of **3b^{IV}** ($\lambda_{em} = 464$ nm) in THF at room temperature.

Figure C9. The absorption and excitation spectra of **3c^I** ($\lambda_{em} = 484$ nm) in THF at room temperature.

2) How do the absorption spectra compare with the computed levels?

Answer: Thank you for your comment. The explicitly spin-adapted TDDFT (X-TDDFT) is performed in order to gain further information of the excited states of radicals (*Theor. Chem. Acc.* **1997**, *96*, 75; *J. Theor. Comput. Chem.* **2003**, *2*, 257). The calculation results show, the energy gap between D₂ and D₁ is higher than 0.46 eV in our current system (except for **3a^{IV}** and **3b^{IV}**) and three times that of known triaryl methyl radical derivatives (for example, *Angew. Chem. Int. Ed.* **2014**, *53*, 11845; *Angew. Chem. Int. Ed.* **2018**, *57*, 2869; *Phys. Chem. Chem. Phys.* **2018**, *20*, 18657), which makes IC slowly and favorable to the emission of anti-Kasha. Relevant discussion has been added in the revised main text (see page 7) and Supporting Information.

3) Since compounds are not emissive in solid state, measurements in blended films could provide additional information (e.g. emission from D₁) on the emissive behavior of these compounds

Answer: Thank you for your comment. According to the your suggestion, thin films of radicals in PMMA were prepared by drop-casting in THF solutions. Our preliminary studies revealed that the doping of radicals into a PMMA matrix produces a strong fluorescence emission in the short-wave range, as observed in their THF solution (Figure C10). These results further confirm that the

emission of radicals comes from highly excited states.

Figure C10. The emission spectra of **3a^I** (left) and **3b^I** (right) with 5% wt in PMMA at room temperature recorded with a 370 nm excitation wavelength.

Reviewers' Comments:

Reviewer #1:

Remarks to the Author:

The new version of the m.s. has been considerably improved with the new included experiments and comments. In this line, data on photostability in THF have been included but there is still a lack of information on the photostability of the radical in a solid matrix since this is the real state for future possible OLED applications. So, these additional data must be included for its final acceptance.

Reviewer #2:

Remarks to the Author:

The authors have addressed my comments and I recommend publication.

Reviewer #3:

Remarks to the Author:

The Authors have adequately replied to my criticism. The Manuscript is now suitable, in my opinion, for publication in Nature Communications.

Response to the Referees' Comments

Reply to the comments by Referee 1

Reviewer #1 (Remarks to the Author):

The new version of the m.s. has been considerably improved with the new included experiments and comments. In this line, data on photostability in THF have been included but there is still a lack of information on the photostability of the radical in a solid matrix since this is the real state for future possible OLED applications. So, these additional data must be included for its final acceptance.

Answer: Thank you for your positive comment. The photostability experiments of the radicals (for **3a^{II}** and **3b^{II}**) dispersed in polymethyl methacrylate (PMMA) film have been added to the Supplementary Information, please see Fig. S68, Page S46.

Reply to the comments by Referee 2

Reviewer #2 (Remarks to the Author):

The authors have addressed my comments and I recommend publication.

Answer: We sincerely thank for the reviewer's positive comments.

Reply to the comments by Referee 3

Reviewer #3 (Remarks to the Author):

The Authors have adequately replied to my criticism. The Manuscript is now suitable, in my opinion, for publication in Nature Communications.

Answer: We sincerely thank for the reviewer's positive comments.